# Adaptive Layer-skipping in Pre-trained LLMs

**Xuan Luo, Weizhi Wang, Xifeng Yan**
Department of Computer Science, UC Santa Barbara
{xuan_luo, weizhiwang, xyan}@cs.ucsb.edu

## Abstract

Various layer-skipping methods have been proposed to accelerate token generation in large language models (LLMs). However, limited attention has been paid to a fundamental question: How do computational demands vary across the generation of different tokens? In this work, we introduce FlexiDepth, a method that dynamically adjusts the number of Transformer layers used in text generation. By incorporating a plug-in router and adapter, FlexiDepth enables adaptive computation in LLMs without modifying their original parameters. Applied to Llama-3-8B, it skips 8 out of 32 layers while maintaining full benchmark performance. Our experiments reveal that computational demands in LLMs significantly vary based on token type. Specifically, generating repetitive tokens or fixed phrases requires fewer layers, whereas producing tokens involving computation or high uncertainty requires more layers. Despite the computational savings, FlexiDepth does not yet achieve wall-clock speedup due to varied skipping patterns and I/O overhead. To inspire future work and advance research on practical speedup, we open-sourced FlexiDepth and a dataset documenting its layer allocation patterns.

| | | |
|---|---|---|
| **Model** | 🤗 | xuan-luo/FlexiDepth-Llama-3-8B-Instruct |
| **Dataset** | 🤗 | xuan-luo/FlexiPatterns-Llama-3-8B-Instruct |

## 1 Introduction

Large language models (LLMs) have achieved remarkable success in various tasks, including translation (Zhu et al., 2024), code generation (Hui et al., 2024), and math reasoning (DeepSeek-AI et al., 2025). Currently, LLMs typically generate each token by performing a full forward pass through all Transformer decoder layers. However, such a uniform allocation is counter-intuitive, as simpler tasks intuitively require fewer computational resources, whereas complex tasks demand more processing. This uniform allocation not only results in computational inefficiencies but may also contribute to overfitting in LLMs. In response to these challenges, various layer-skipping techniques have been proposed. One line of research leverages statistical information, such as the difference between layer inputs and outputs, to identify and skip less important layers (Men et al., 2024; He et al., 2024b; Zhang et al., 2024). Another collection of methods involves early-exit (Varshney et al., 2024; Jazbec et al., 2024; Schuster et al., 2022), where a confidence measure or a router dynamically determines whether to bypass all subsequent layers at an intermediate point of LLM. While these techniques have successfully reduced computational costs, limited attention has been paid to a fundamental question: How do computational demands vary across the generation of different tokens?

In this work, we propose FlexiDepth, a method for adaptive layer-skipping in pre-trained large language models. At each transformer layer, our method determines whether the hidden state input shall forward pass the layer or skip it. This layer-wise approach offers the flexibility to tailor the computation path for each token, enabling processing with varying numbers of transformer layers.

[Context] The grey seal was found at Cruden Bay on Tuesday. The Scottish SPCA removed the seal from beach and took it to the charity's rescue centre... Thankfully there were no injuries in this case and we were able to release him relatively quickly.

[Prompt] Where the grey seal was found?

**The grey seal was found at Cruden Bay.**

[Prompt] Please summarize this paragraph into a single sentence:

**A grey seal was found tangled in netting at Cruden Bay , rescued**
**by the Scottish S PCA , and released back into the water with no injuries.**

[Prompt] Please calculate the sum of the 8 numbers in the list: [99, 45, 12, 78, 33, 66, 21, 54]

**To calculate the sum of all 8 numbers in the list, I'll add each number one by one:**
**99 + 45 = 144**
**144 + 12 = 156**
**156 + 78 = 234**
**234 + 33 = 267**
**267 + 66 = 333**
**333 + 21 = 354**
**354 + 54 = 408**

**The sum of all 8 numbers in the list is 408.**

Figure 1: DepthMap illustrating layer-skipping patterns when applying FlexiDepth to Llama-3-8B-Instruct. The light-to-dark blue gradient indicates layer usage ranging from 16 layers to 32 layers.

Contrast to layer skipping methods requiring training LLM from scratch, i.e. mixture-of-depth (MoD) (Raposo et al., 2024), FlexiDepth enables adaptive layer-skipping in pre-trained LLMs without modifying their original parameters. Specifically, FlexiDepth introduces two light-weight plug-in modules at each decoder layer: (1) a router that makes binary decisions about whether to skip the layer, and (2) a adapter that resolves representation mis-alignments caused by layer skipping. The router and adapter are the only trainable components within our framework. All other parameters are inherited from a pre-trained LLM and remain frozen. At each layer, the input hidden states are first evaluated by the router and directed to go through or skip the layer. For hidden states that go through the layer, they are processed by the original attention and FFN modules of the LLM. Conversely, hidden states that skip the layer are instead processed by the adapter. The adapter's role is to transform these skipped hidden states, aligning their representation to match those that undergo full processing. Finally, both the processed hidden states and the adapted hidden states are combined to produce the layer output. To optimize the router and adapter, we introduce a layer skipping loss that works in conjunction with the original next-token prediction loss. This loss function encourages layer skipping by penalizing the number of layers used for generation, while maintaining generation quality through the prediction loss. After training, the model can dynamically adjust its layer usage based on the input.

We comprehensively evaluate FlexiDepth across a diverse set of benchmarks. When applied to Llama-3-8B-Instruct (Dubey et al., 2024), FlexiDepth preserves full performance (100.7%) while skipping 8 out of 32 layers, significantly outperforming existing layer-skipping methods, particularly in continuous generation tasks.

FlexiDepth also reveals how computational demands vary when generating different types of tokens. For demonstration, we created a colored map that shows the number of layers used to generate each token, which we call 'DepthMap'. Figure 1 (left) shows that summarization typically requires more layers on average than extractive question answering. Similarly, Figure 1 (right) reveals that in mathematical reasoning tasks like addition, tokens on the left-hand side of equations need fewer layers to generate than those on the right-hand side. This type of adaptive allocation appears to resonate with human intuition. For instance, humans typically exert more effort in summarization tasks to condense the entire context into a concise form, while finding extractive question answering easier as it only requires retrieval and extraction. Likewise, in mathematical tasks, deriving the right-hand side of an equation involves computation that consumes more steps. Shifting from token-specific patterns to overall layer usage, we observe that layer utilization follows a long-tail distribution, with early and final layers used more frequently than middle layers. To inspire further research along this direction, we compile a dataset documenting the layer allocation behaviors of FlexiDepth across various tasks.

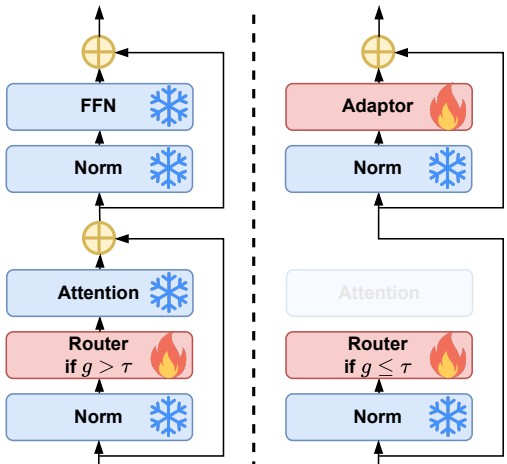

Figure 2: The FlexiDepth layer. Left: Full-processing path where hidden states undergo the pre-trained attention and FFN modules. Right: Skipping path where hidden states bypass the attention module and processed by a lightweight adapter. The router and adaptor (in red) are the only trainable components.

## 2 Method

In this section, we introduce FlexiDepth, a framework for adaptive layer-skipping in pre-trained LLMs. Each FlexiDepth layer augments a standard decoder layer with a lightweight router and adapter, keeping the original parameters frozen. The router evaluates normalized input hidden states to compute the gating score $g$. Based on a threshold $\tau$, hidden states with $g > \tau$ are routed to the full-processing path (Figure 2, left), where they undergo the pre-trained attention and FFN modules. Hidden states with $g \leq \tau$ are routed to the skipping path (Figure 2, right), where they are instead processed by a lightweight adapter. In the following parts, we elaborate on the core designs of FlexiDepth, including routing, attention skipping, FFN skipping, and layer-skipping loss.

### 2.1 Routing

FlexiDepth employs a router at each layer to compute gating scores for routing. Let $X = [x_1, x_2, \ldots, x_T] \in \mathbb{R}^{T \times d}$ denote the input hidden states of the current layer, where $T$ is the sequence length and $d$ is the hidden dimension. The router computes their corresponding gating scores $G = [g_1, g_2, \ldots, g_T] \in \mathbb{R}^T$ as follows:

$$G = \sigma(\text{Router}(\text{Norm}(X))), \tag{1}$$

where Norm refers to the RMSNorm (Zhang & Sennrich, 2019), and $\sigma$ is the sigmoid function to ensure $g_i \in (0, 1)$. In previous works (Raposo et al., 2024; Tan et al., 2024), the router is typically implemented as a simple linear transformation, and jointly optimized with the transformer during training. However, in our setting, the original parameters of the pre-trained transformer are frozen. Relying on a linear transformation is insufficient to capture the nuances of the hidden states for routing. To address this issue, we design a parameter-efficient router based on a bottlenecked MLP:

$$\text{Router}(z) = W_{\text{r}} \cdot (W_{\uparrow} \cdot \text{Norm}(\tanh(W_{\downarrow} z))), \tag{2}$$

where $W_{\downarrow} \in \mathbb{R}^{d_r \times d}$, $W_{\uparrow} \in \mathbb{R}^{d \times d_r}$, and $W_{\text{r}} \in \mathbb{R}^{1 \times d}$ represent the weights of down-projection, up-projection, and router head, respectively. The $d_r$ denotes the bottleneck dimension.

After computing the gating scores $G$, we apply a threshold-based routing mechanism using a predefined threshold $\tau$. For each hidden state $x_i$, if $g_i > \tau$, it is routed to the full-processing path (Figure 2, left), where the output is scaled by $g_i$ to ensure gradient flow to the router.

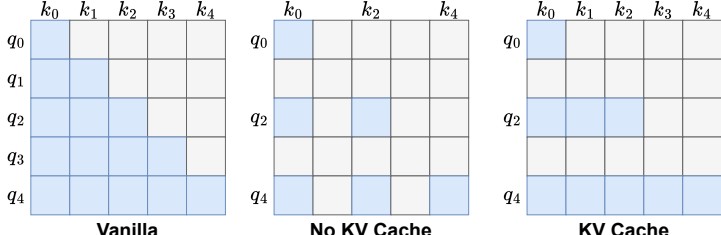

Figure 3: Comparison of attention masks. Left: the vanilla attention mask. Middle: the attention mask without KV Cache, where hidden states $x_1$ and $x_3$ skip this layer. Right: the attention mask with KV Cache, where despite $x_1$ and $x_3$ skipping this layer, their corresponding keys and values are still computed.

If $g_i \leq \tau$, the hidden state is routed to the skipping path (Figure 2, right), with the output scaled by $(1 - g_i)$. This process is formalized as:

$$x_i' = \begin{cases} g_i \cdot \text{FFN}(\text{Norm}(\text{Attn}(\text{Norm}(x_i)) + x_i)) + \text{Attn}(\text{Norm}(x_i)) + x_i, & \text{if } g_i > \tau \\ (1 - g_i) \cdot \text{Adapter}(\text{Norm}(x_i)) + x_i, & \text{if } g_i \leq \tau \end{cases} \quad (3)$$

where $x_i'$ is the output hidden state for token $i$, Norm represents layer normalization, FFN denotes the feed-forward network, and Adapter$(\cdot)$ is the lightweight adapter.

## 2.2 Attention skipping

In the skipping path of FlexiDepth (Figure 2, right), the input hidden states simply bypass the attention module via a shortcut. However, this straightforward approach can lead to a significant loss of contextual information, resulting in performance degradation. In autoregressive generation, the attention mechanism computes query, key, and value vectors for each token, where keys and values are cached (KV Cache) (Pope et al., 2023) to avoid redundant computation in subsequent decoding steps. When hidden states skip the attention module, their corresponding key and value vectors are not generated, preventing subsequent tokens from attending to them. This issue is illustrated in Figure 3 (middle), where in this example, hidden states $x_1$ and $x_3$ are routed to skip the layer. Naively skipping the attention module for these tokens results in missing key-value pairs (i.e., $k_1, v_1$ and $k_3, v_3$). Therefore, the subsequent tokens, such as $q_4$, can no longer attend to them.

To address this issue, we propose a straightforward solution: we continue computing the keys and values for hidden states that skip the layer. Specifically, while we can omit the query vectors (e.g., $q_1, q_3$) and their associated scaled dot-product operations, we retain the calculations of the corresponding keys ($k_1, k_3$) and values ($v_1, v_3$). This approach, as demonstrated in Figure 3 (right), ensures that future tokens maintain access to the full set of contextual information, preserving the integrity of autoregressive generation.

## 2.3 FFN skipping

We observe that directly skipping the feedforward network (FFN) via a simple shortcut significantly degrades model performance. Unlike the attention module, which only involves linear transformations, the FFN module introduces nonlinearities. Consequently, hidden states transformed by the FFN may not share the same latent space as those that skip it. To mitigate this issue, we employ a lightweight adapter to align their representations. This adapter follows the same structure as the FFN but features a significantly reduced intermediate dimension. As shown in Figure 2 (right), the adapter is placed at the same position as the skipped FFN.

## 2.4 Layer-skipping loss

To balance computational efficiency and generation quality, we introduce a skipping loss that is jointly optimized with the next-token prediction loss. This loss is designed to penalize

the squared sum of the layers used. The squared loss imposes a larger penalty on tokens that activate more layers and a smaller penalty on those that use fewer. This non-uniform penalty stabilizes training by preventing the model from falling into extreme patterns, such as skipping all layers or none. The loss function is formulated as:

$$\mathcal{L}_{skip} = \frac{1}{T} \sum_{t=1}^{T} \left( \sum_{l=1}^{L} g_t^l \right)^2, \tag{4}$$

where $g_t^l$ denotes the gating score for layer $l$ at time step $t$. The final loss integrates the skipping loss $\mathcal{L}_{skip}$ with the original language modeling loss $\mathcal{L}_{lm}$, weighted by a factor $\alpha$:

$$\mathcal{L} = \alpha \cdot \mathcal{L}_{skip} + \mathcal{L}_{lm}. \tag{5}$$

## 3 Experiments

### 3.1 Implementation details

We implement FlexiDepth on the pre-trained Llama-3-8B-Instruct model (Dubey et al., 2024), which consists of 32 transformer layers. To enable adaptive layer-skipping, we convert the latter 16 layers into FlexiDepth layers, as prior works (Men et al., 2024; Sun et al., 2024) and empirical studies suggest that skipping earlier layers degrades performance. For each FlexiDepth layer, the router, as defined in Equation 2, uses a bottleneck dimension $d_r = \frac{1}{16} d$, where $d$ is the hidden dimension. The router's gating function is implemented using SparseMixer (Liu et al., 2023a;b) to ensure differentiability. The adapter has the same structure as the original FFN but reduces the intermediate dimension by a factor of 16. For the loss function in Equation 5, we set the coefficient $\alpha = 1 \times 10^{-3}$, which results in approximately 8 layers skipped during generation. We train FlexiDepth on the Tulu-v2 dataset (Ivison et al., 2023) for 3 epochs using the AdamW (Loshchilov & Hutter, 2019) optimizer with a learning rate of $1 \times 10^{-4}$, $\beta_1 = 0.9$, $\beta_2 = 0.999$, and $\epsilon = 1 \times 10^{-8}$. We use a warmup ratio of 0.03 and a global batch size of 64. The training takes approximately 7 hours on 8 NVIDIA A100-PCIE-40GB GPUs.

### 3.2 Main results

In this part, we present the experimental results of applying the proposed FlexiDepth to the pre-trained Llama-3-8B-Instruct model (Dubey et al., 2024).

**Benchmarks.** We evaluate FlexiDepth across a diverse set of tasks. For single-token generation benchmarks, we include: MMLU (Hendrycks et al., 2021), HellaSwag (Zellers et al., 2019), and Winogrande (Sakaguchi et al., 2020). For multi-token generation tasks, we test on: GSM8K (Cobbe et al., 2021), HumanEval (Chen et al., 2021), CoQA (Reddy et al., 2019). Evaluations are conducted using the lm-evaluation-harness toolkit (Gao et al., 2024), with 5-shot settings for MMLU, HellaSwag, Winogrande, and GSM8K, and zero-shot settings for HumanEval and CoQA. The evaluation metrics are accuracy (acc) for MMLU, normalized accuracy (acc_norm) for HellaSwag, accuracy (acc) for Winogrande, exact-match (EM) for GSM8K, pass-at-1 (Pass@1) for HumanEval, and F1 score (F1) for CoQA.

**Baselines.** We compared FlexiDepth with previous layer-skipping methods that are compatible with LLMs. All baseline methods are applied to the LLama-3-8B-Instruct model, which consists of 32 transformer layers. We define $k$ as the number of layers to skip during generation, with all methods configured to skip the same number of layers for fair comparison. The baseline methods are as follows: (1) LayerSkip (Elhoushi et al., 2024): This early-exit method skips the last $k$ consecutive layers during decoding and employs speculative decoding (Leviathan et al., 2023) to refine generation results. For comparison, we disable speculative decoding. (2) ShortGPT (Men et al., 2024): This approach prunes $k$

| Methods | Single-Token Generation | | | Multi-Token Generation | | | Retain % |
|---|---|---|---|---|---|---|---|
| | MMLU | Hellaswag | Winogrande | GSM8K | HumanEval | CoQA | |
| Vanilla | 0.673 | 0.706 | 0.744 | 0.679 | 0.299 | 0.784 | 100.0% |
| Skip 4 Layers | | | | | | | |
| LayerSkip | 0.659 | 0.636 | 0.676 | 0.004 | 0.0 | 0.350 | 54.0% |
| ShortGPT | 0.664 | 0.662 | 0.700 | 0.536 | 0.092 | 0.145 | 69.1% |
| LaCo | 0.671 | 0.693 | 0.724 | 0.581 | 0.031 | 0.778 | 81.7% |
| MindSkip | 0.664 | 0.698 | 0.722 | 0.378 | 0.189 | 0.720 | 84.2% |
| Ours | 0.663 | 0.724 | 0.756 | 0.695 | 0.390 | 0.810 | 106.5% |
| Skip 8 Layers | | | | | | | |
| LayerSkip | 0.650 | 0.525 | 0.640 | 0.0 | 0.0 | 0.049 | 43.9% |
| ShortGPT | 0.307 | 0.462 | 0.597 | 0.001 | 0.0 | 0.005 | 32.0% |
| LaCo | 0.656 | 0.628 | 0.695 | 0.065 | 0.006 | 0.707 | 65.3% |
| MindSkip | 0.602 | 0.650 | 0.646 | 0.039 | 0.024 | 0.620 | 60.2% |
| Ours | 0.616 | 0.705 | 0.735 | 0.662 | 0.341 | 0.801 | 100.7% |

Table 1: Performance comparison based on Llama-3-8B-Instruct, which consists of 32 layers. Retain % represents the percentage of average retained benchmark performance.

| Model | MMLU | Hellaswag | Winogrande | GSM8K | HumanEval | CoQA | Retain % |
|---|---|---|---|---|---|---|---|
| Llama-2-13B | 0.492 | 0.727 | 0.714 | 0.236 | 0.102 | 0.790 | 100.0% |
| FlexiDepth | 0.481 | 0.741 | 0.716 | 0.253 | 0.096 | 0.780 | 100.2% |
| Skipped | 7.08 | 1.98 | 5.00 | 6.65 | 11.61 | 6.19 | - |
| Llama-3-8B | 0.673 | 0.706 | 0.744 | 0.679 | 0.299 | 0.784 | 100.0% |
| FlexiDepth | 0.663 | 0.743 | 0.756 | 0.657 | 0.323 | 0.803 | 102.1% |
| Skipped | 4.12 | 2.00 | 3.97 | 10.42 | 9.55 | 7.44 | - |
| Qwen-2.5-3B | 0.651 | 0.702 | 0.639 | 0.576 | 0.487 | 0.705 | 100.0% |
| FlexiDepth | 0.643 | 0.701 | 0.679 | 0.583 | 0.476 | 0.741 | 101.5% |
| Skipped | 1.23 | 1.25 | 1.15 | 1.71 | 1.69 | 1.88 | - |

Table 2: Performance and skipping patterns of FlexiDepth on instruction-tuned models. Each model includes baseline performance (first row), FlexiDepth's performance (second row), and average skipped layers (third row). The '-Instruct' are omitted for simplicity.

layers deemed less important by assessing their input-output difference. (3) LaCo (Yang et al., 2024b): This method employs a layer-collapse strategy to reduce the model by $k$ layers, merging subsequent layers into a prior one using a Reserving-Differences-while-Seeking-Common (RDSC) approach. (4) MindSkip (He et al., 2024a): This method explores attention, FFN, and layer skipping via a simple linear router, finding that skipping FFN or entire layers leads to significant performance degradation. For comparison, we employ its layer skipping setting and train it on the same tulu-v2 (Ivison et al., 2023) dataset.

FlexiDepth demonstrates a varied number of skipped layers for different tasks. To compare fairly with baselines at a consistent number of skipped layers $k$, we trained multiple FlexiDepth models with varying penalty weights $\alpha$ and evaluated their performance. For each task, we report the results of the model that achieves the target number of skipped layers $k$. Later, we will highlight FlexiDepth's adaptability by showing its performance across these tasks with a single, fixed $\alpha$.

Table 1 presents the comparison results. FlexiDepth consistently outperforms baseline methods, especially in multi-token generation tasks. For example, when skipping 8 layers, baseline methods retain reasonable performance on single-token generation benchmarks and the short-form question-answering task CoQA. However, they suffer from near-zero accuracy on GSM8K and HumanEval that requires longer reasoning. In contrast, FlexiDepth excels across both single-token and multi-token generation tasks, achieving an average performance retention of 100.7%.

Interestingly, FlexiDepth even slightly surpasses the original model's performance on certain benchmarks, highlighting the potential benefits of adaptive depth. To further examine whether this improvement stems from training data or the mechanism itself, we compare FlexiDepth with the fully fine-tuned allenai/llama-3-tulu-2-8b model on huggingface, which is trained on the same dataset as FlexiDepth. Notably, this model achieves 0.554 on GSM8K and 0.354 on HumanEval, while FlexiDepth achieves 0.695 and 0.390 on the same tasks—even when skipping 4 layers. This result suggests that the performance gains are not solely attributable to the training data. We hypothesize that its adaptive layer-skipping mechanism may implicitly act as a form of regularization, by skipping less informative or noisy parameters during inference, thus enhancing generalization. Future work is needed to explore this possibility and investigate the broader potential of adaptive layer-skipping.

## 3.3 FlexiDepth on different language models

We evaluate FlexiDepth on language models of varying sizes. These models are trained under the identical configuration and dataset described in our implementation details. Larger models, specifically Llama-2-13B-Instruct and Llama-3-8B-Instruct, exhibit greater number of layers skipped compared to the smaller Qwen-2.5-3B-Instruct (Yang et al., 2024a). Specifically, Llama-2-13B-Instruct and Llama-3-8B-Instruct skip an average of 6.42 and 6.25 layers, respectively, across benchmarks, while Qwen-2.5-3B-Instruct skips only about 1.49 layers. This trend indicates that larger models possess greater inherent redundancy, enabling more aggressive layer skipping without notable performance degradation. Given this trend, FlexiDepth can potentially be applied to even larger models to further exploit their redundancy. For instance, it could be applied to large mixture-of-experts models like DeepSeek-V3 (DeepSeek-AI et al., 2024), where experts are distributed across multiple servers during inference. By skipping entire layers, FlexiDepth could reduce cross-server communication overhead, enhancing inference efficiency of MoE models.

## 3.4 Layer allocation dataset

To investigate these layer-skipping patterns and their relationship to task complexity, we constructed a dataset using FlexiDepth with Llama-3-8B-Instruct as the base model. This dataset captures layer usage for token generation in two key domains: language modeling and math reasoning. By recording the number of layers utilized per token, we aim to uncover how FlexiDepth dynamically adjusts the layers to meet varying task demands.

**Text Generation.** For text generation tasks, we constructed a dataset by randomly sampling 100 paragraphs from the XSum test set (Narayan et al., 2018), which is a collection of news articles. We evaluated FlexiDepth on three subtasks for each paragraph: copying, summarization, and continuation:

```
Please copy this paragraph: <paragraph>...</paragraph>
Directly output the copied paragraph here:
-----------------------------------------------------------------------------
Please summarize this paragraph into a single sentence:
<paragraph>...</paragraph>
Directly output the summarized paragraph here:
-----------------------------------------------------------------------------
Please continue writing this paragraph: <paragraph>...</paragraph>
Directly output the continued paragraph here:
```

We recorded the number of layers used per token across all outputs: copying averaged 21.95 layers (variance 6.05), summarization averaged 28.65 layers (variance 18.31), and continuation averaged 30.27 layers (variance 12.59). These results reveal a clear pattern: tasks requiring deeper contextual understanding, such as continuation and summarization, utilize more layers compared to the simpler copying task. The higher variance in summarization (18.31) and continuation (12.59) versus copying (6.05) suggests greater variability in layer allocation, likely reflecting the diverse cognitive demands of generating concise summaries or creative continuations. Detailed examples of these patterns are presented in Appendix A.

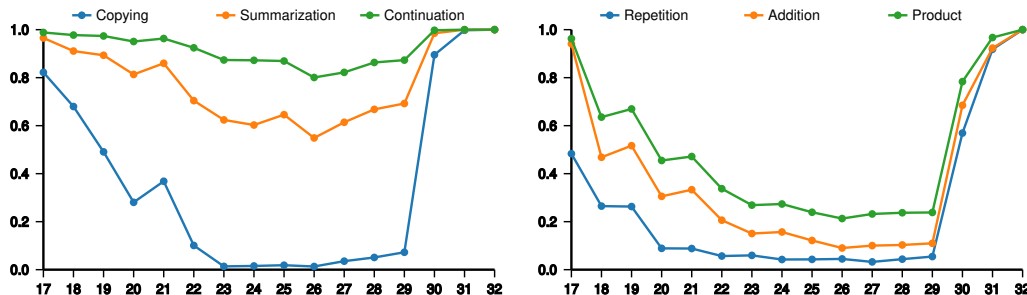

Figure 4: Percentage of tokens processed by transformer layers 17 to 32. The x-axis represents the layer index, and the y-axis represents the percentage of tokens processed by the layer.

**Math Reasoning.**    For math reasoning, we created a dataset of 100 samples, each featuring a randomly generated list of 5–10 integers between 10 and 99. This controlled setup enables us to test basic arithmetic operations systematically. FlexiDepth was prompted to process each list with three subtasks: repeating the list, calculating the sum, and the product.

```
Please repeat the following list for 5 times: [...]
------------------------------------------------------------------------
Please calculate the sum of numbers in the following list: [...]
------------------------------------------------------------------------
Please calculate the product of numbers in the following list: [...]
```

The layer usage pattern across these three tasks was as follows: repeating averaged 20.09 layers (variance 8.08), addition averaged 22.45 layers (variance 14.67), and multiplication averaged 23.90 layers (variance 21.61). Appendix A provides additional results, showing that tokens tied to arithmetic operations consistently require more layers than directly copying numbers from the context.

## 3.5   Layer utilization pattern

To complement the token-centric analysis in prior subsections, we examine the overall layer utilization pattern of FlexiDepth on Llama-3-8B-Instruct. We measure the percentage of tokens processed by each transformer layer across the tasks outlined earlier: text generation (copying, summarization, continuation) and math reasoning (repeating, addition, product). Figure 4 visualizes the results, with the left plot showing language modeling tasks and the right plot showing math reasoning tasks. Layer utilization exhibits a bowl-like pattern, with higher token processing at the initial and final layers, and reduced usage in the middle layers. For language tasks, copying exhibits the deepest dip in middle-layer usage, while continuation utilizes the most middle layers, reflecting varying task complexity. Similarly, in math reasoning, repeating shows the least middle-layer usage, whereas product requires the most, which is consistent with increasing task difficulty. The overall layer usage distribution approximately follows a long-tail pattern, where most tokens are processed by a few layers, and others are selectively skipped. We hypothesize that early layers primarily parse and understand input context, middle layers perform task-specific processing that varies with complexity, and final layers focus on decoding for coherent output generation. Future studies are needed to verify the specialized roles of different layers.

## 3.6   Ablation Studies

We present ablation studies in this section. All models are based on Llama-3-8B-Instruct and trained using identical configurations as described in our implementation details.

**Linear Router.**    In FlexiDepth, we employed a bottlenecked MLP as the router to facilitate nuanced routing decisions. When replacing this component with a simpler linear layer

|  | MMLU | Hellaswag | Winogrande | GSM8K | HumanEval | CoQA | Retain % |
|---|---|---|---|---|---|---|---|
| Vanilla | 0.673 | 0.706 | 0.744 | 0.679 | 0.299 | 0.784 | 100.0% |
| FlexiDepth | 0.663 | 0.743 | 0.756 | 0.657 | 0.323 | 0.803 | 102.1% |
| Linear Router | 0.619 | 0.684 | 0.701 | 0.131 | 0.055 | 0.716 | 68.7% |
| No KV Cache | 0.525 | 0.699 | 0.745 | 0.366 | 0.226 | 0.775 | 84.3% |
| No Adapter | 0.226 | 0.356 | 0.579 | 0.004 | 0.0 | 0.047 | 28.1% |

Table 3: Ablation studies based on Llama-3-8B-Instruct.

|  | MMLU | Hellaswag | Winogrande | GSM8K | HumanEval | CoQA | Retain % |
|---|---|---|---|---|---|---|---|
| Vanilla | 0.673 | 0.706 | 0.744 | 0.679 | 0.299 | 0.784 | 100.0% |
| $\alpha = 5e-4$ | 0.671 | 0.730 | 0.740 | 0.687 | 0.347 | 0.806 | 103.7% |
| Skipped | 2.32 | 0.98 | 1.03 | 3.49 | 6.40 | 4.01 | - |
| $\alpha = 1e-3$ | 0.663 | 0.743 | 0.756 | 0.657 | 0.323 | 0.803 | 102.1% |
| Skipped | 4.12 | 2.00 | 3.97 | 10.42 | 9.55 | 7.44 | - |
| $\alpha = 2e-3$ | 0.637 | 0.721 | 0.736 | 0.528 | 0.140 | 0.778 | 86.6% |
| Skipped | 6.10 | 3.85 | 5.60 | 12.53 | 11.88 | 9.92 | - |

Table 4: The influence of the coefficient $\alpha$ for layer skipping.

followed by a sigmoid activation, as previously used in mixture-of-depth (MoD) (Raposo et al., 2024), we observe a substantial decrease in performance, particularly on the math reasoning task. Our analysis shows that the linear router mistakenly assigns too few layers to tokens requiring complex computations, significantly reducing the GSM8K performance from 0.657 to 0.131. Overall, the linear router retains only 68.7% of the original performance, emphasizing the importance of the bottlenecked MLP for routing.

**No KV Cache.** We conducted an ablation study in which the KV cache is not computed for tokens routed to skip layers (illustrated as "No KV Cache" in Figure 3). Removing the KV cache results in substantial performance decline, retaining only 84.3% of the original performance. This highlights the need to retain contextual information during generation.

**No adapter.** Many layer skipping methods do not have adapters. We therefore investigate the performance of FlexiDepth without adapter. This simplification drastically reduces model performance to merely 28.1% of the original FlexiDepth results. The adapter proves critical in aligning latent representations between skipped and processed states, ensuring representation coherence across the model's layers.

**Different Coefficient.** We investigate the impact of the coefficient $\alpha$ in the layer skipping loss (Equation 5). With $\alpha = 5e-4$, the model only skips an average of 3.04 layers. Increasing to $\alpha = 2e-3$ results in more layers being skipped (averaging 8.31 layers) but causes significant performance drops on reasoning tasks, with GSM8K decreasing to 0.528 and HumanEval to 0.140. The default value $\alpha = 1e-3$ achieves a better balance.

## 4 Limitations

Although FlexiDepth reduces FLOPs by adaptively skipping transformer layers, our implementation does not lead to improved throughput on the existing GPU hardware (see Appendix B). In FlexiDepth, samples within the same batch can take different execution paths during decoding. Consequently, each FlexiDepth layer must handle tokens that undergo full processing alongside those that skip layers, requiring simultaneous management of both computation and skip branches. This introduces overhead from control-flow management and irregular memory accesses, outweighing the speedup of theoretical FLOP reductions. We are currently exploring approaches to achieve actual throughput improvements with

FlexiDepth and have found that its framework can be adapted for early exit strategies (see Appendix C). Future work could also investigate hardware-oriented optimizations to address these bottlenecks, leveraging techniques such as token grouping (Rajbhandari et al., 2022), expert sharding (Rajbhandari et al., 2022), and load balancing (Huang et al., 2024) to better align with GPU architectures. Such hardware-tailored enhancements could further unlock the efficiency potential of FlexiDepth in practice.

## 5 Related works

The existing language models can be broadly classified into three categories: encoder-only models (e.g., BERT (Devlin et al., 2019)), encoder-decoder models (e.g., T5 (Raffel et al., 2020)), and decoder-only models (e.g., GPT (Brown et al., 2020)). Across these architectures, various conditional computation methods have been proposed to optimize resource allocation by dynamically processing tokens.

In encoder-only models, conditional token processing has been explored to prune less critical tokens during processing. For instance, PoWER-BERT (Goyal et al., 2020) and LTP Kim et al. (2022) utilize attention scores to assess token importance, dynamically eliminating those deemed less significant. Similarly, LoT (Kim et al., 2023) introduces a router at each layer to determine whether tokens should be processed or skipped, tailoring computation to token relevance. These approaches leverage the bidirectional nature of encoders, allowing noncausal routing decisions based on full sequence context.

For encoder-decoder models, methods such as CoDA (Lei et al., 2023) and COLT5 (Ainslie et al., 2023) enable conditional computation within the encoder component. By replacing standard attention or feed-forward network (FFN) modules with lightweight alternatives for certain tokens, these techniques assign varying computational resources based on token complexity. However, their noncausal design constrains their applicability to decoder.

In decoder-only models, the autoregressive nature poses unique challenges for dynamic token processing, as tokens must be generated sequentially while preserving contextual dependencies. Piorneering efforts like MoD (Raposo et al., 2024) and SkipLayer (Zeng et al., 2023) address this by deploying a router at each Transformer layer, enabling tokens to dynamically bypass it. These models are trained end-to-end from scratch, integrating the routing mechanism into the learning process. Duo-LLM (Alizadeh et al., 2024) also employs a router for conditional computation, but with a distinct method. It first constructs a dataset of oracle routing strategies via exhaustive enumeration and then trains the router on this data. MindSkip (He et al., 2024a) enables skipping in pre-trained models via learnable routers. It finds that bypassing FFN modules or entire layers significantly degrades performance and instead proposes a method to skip the attention module efficiently. Unlike these approaches, our method, FlexiDepth, enables adaptive layer-skipping in pre-trained models without compromising generation performance by using a plug-in router and adapter architecture.

## 6 Conclusion

We present FlexiDepth, an approach to dynamically skip layers in pre-trained large language models. It achieves state-of-the-art performance without altering the original model parameters. FlexiDepth offers valuable insights into the varying computational demands of token generation. We constructed a dataset capturing these allocation patterns to encourage further study of computational needs of tokens, thereby advancing the development of more adaptive and resource-efficient language models.

## Acknowledgements

This work was partially supported by the BioPACIFIC Materials Innovation Platform of the National Science Foundation under Award No. DMR-1933487. We would like to thank Facebook (now Meta) for donating the A100-40G GPUs for conducting the experiments.

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

# A   Layer allocation Examples

In this section, we present examples to illustrate the layer allocation patterns of FlexiDepth, using the pre-trained Llama-3-8B-Instruct model (Dubey et al., 2024) as the base.

## A.1   Language Task

**Copy:**   Here is an example where FlexiDepth uses 21.84 layers in average per token:

Please copy this paragraph: <paragraph>Bissouma penned a professional deal with the northern club only in July 2016, four months after joining the reserves from AS Real Bamako. The 20-year-old, whose previous deal ran until June 2019, says he is very happy at Stade Pierre-Mauroy. "In football, when you put in the work you always get rewarded," Bissouma told the club's website. "I still have a lot of progress to make. I want to give myself every chance of doing that and fulfilling my objectives. "The first is to keep the club in Ligue 1, and then I would also like to win a trophy with Lille." Bissouma has scored once in 17 appearances for the club since making his debut against Toulouse in September 2016.</paragraph> Directly output the copied paragraph here:

**Bissouma penned a professional deal with the northern club only in July 2016, four months after joining the reserves from AS Real Bamako.\nThe 20–year–old, whose previous deal ran until June 2019, says he is very happy at Stade Pierre–Mauroy .\n"In football, when you put in the work you always get rewarded," Bissouma told the club's website.\n"I still have a lot of progress to make. I want to give myself every chance of doing that and fulfilling my objectives.\n"The first is to keep the club in Ligue 1, and then I would also like to win a trophy with Lille."\nBissouma has scored one in 17 appearances for the club since making his debut against T oulouse in September 2016.<|eot_id|>**

This task requires minimal contextual understanding, as it involves direct reproduction, which explains the relatively low layer usage. The first token 'B' requires much more layers to generate because the generation of the first token involves the prefilling stage, where the whole context is been processed.

**Summarization:**   Here is an example using 29.15 layers in average per token:

Please summarize this paragraph into a single sentence: <paragraph>Bissouma penned a professional deal with the northern club only in July 2016, four months after joining the reserves from AS Real Bamako. The 20-year-old, whose previous deal ran until June 2019, says he is very happy at Stade Pierre-Mauroy. "In football, when you put in the work you always get rewarded," Bissouma told the club's website. "I still have a lot of progress to make. I want to give myself every chance of doing that and fulfilling my objectives. "The first is to keep the club in Ligue 1, and then I would also like to win a trophy with Lille." Bissouma has scored once in 17 appearances for the club since making his debut against Toulouse in September 2016.</paragraph> Directly output the summarized paragraph here:

**Bissouma, who signed a professional deal with Lille in July 2016 and has scored once in 17 appearances, is happy at the club and aims to keep them in Ligue 1 and win a trophy.<|eot_id|>**

Summarization involves understanding and condensing the input into a concise form, requiring deeper contextual abstraction and rephrasing, which leads to a higher layer usage of 29.15 compared to copying. We observe that tokens forming predictable collocations, such as "souma" following "Bis" in "Bissouma" or "trophy" after "win a" in "win a trophy," require fewer layers due to their low uncertainty after the initial tokens are generated.

**Continuation:**   Here is an example that utilizes 31.14 layers in average:

Please continue writing this paragraph: <paragraph>Bissouma penned a professional deal with the northern club only in July 2016, four months after joining the reserves from AS Real

Bamako. The 20-year-old, whose previous deal ran until June 2019, says he is very happy at Stade Pierre-Mauroy. "In football, when you put in the work you always get rewarded," Bissouma told the club's website. "I still have a lot of progress to make. I want to give myself every chance of doing that and fulfilling my objectives. "The first is to keep the club in Ligue 1, and then I would also like to win a trophy with Lille." Bissouma has scored once in 17 appearances for the club since making his debut against Toulouse in September 2016.</paragraph> Directly output the continued paragraph here:

**Despite his young age, Bissouma has already shown impressive maturity in his performances, with his manager, Marcel Blondeau, praising him for his ability to adapt to the team's system. "He is a very intelligent player," Blondeau said. "He can play in different positions and has a good understanding of the game. He is also very strong in the tackle and can win the ball back." With his contract running until 2021, Bissouma is expected to continue playing a crucial role in Lille's midfield for the coming seasons.<|eot_id|>**

Continuation demands creative generation and the maintenance of contextual coherence, necessitating long-range dependencies and novel content creation, which results in a higher layer usage of 31.14. The generation process typically engages more layers, except for predictable tokens like names ("Bissouma" and "Blondeau"), which require fewer layers due to their memorized nature and low uncertainty.

## A.2 Math Task

**Repeating:** Here is an example that uses 20.15 layers in average:

Please repeat the following list for 5 times: [63, 15, 43, 75, 72, 61, 48, 71]

**Here is the list repeated 5 times**

**[63, 15, 43, 75, 72, 61, 48, 71]**

**[63, 15, 43, 75, 72, 61, 48, 71]**

**[63, 15, 43, 75, 72, 61, 48, 71]**

**[63, 15, 43, 75, 72, 61, 48, 71]**

**[63, 15, 43, 75, 72, 61, 48, 71]<|eot_id|>**

Repeating is a memory-based task requiring minimal computation, as it involves recalling and reproducing the input list, which explains the low layer usage. This example fits within the reported variance (8.08), reflecting the task's consistent layer needs.

**Addition:** Here is an example that uses 22.31 layers in average:

Please calculate the sum of numbers in the following list: [63, 15, 43, 75, 72, 61, 48, 71]

**Let's calculate the sum of the numbers in the list**

**63 + 15 = 78**

**78 + 43 = 121**

**121 + 75 = 196**

**196 + 72 = 268**

**268 + 61 = 329**

**329 + 48 = 377**

**377 + 71 = 448**

**The sum of the numbers in the list is 448<|eot_id|>**

Addition requires iterative computation, using more layers than repeating since FlexiDepth calculates each intermediate sum. The right-hand side of each equation (e.g., "78", "121")

| Method | MMLU | Hellaswag | Winogrande | GSM8K | HumanEval | CoQA |
|---|---|---|---|---|---|---|
| Vanilla | 32.21 | 11.19 | 26.55 | 5.53 | 10.34 | 8.29 |
| FlexiDepth | 33.43 | 11.20 | 26.17 | 5.46 | 10.18 | 7.78 |

Table 5: Throughput (iterations/s) comparison across benchmarks.

demands more layers due to the transformer's internal computation, while the left-hand side (e.g., "63 + 15") generally uses fewer layers as it copies numbers from the input. We also find that the initial tokens on the left-hand side, such as "63" in "63 + 15", use slightly more layers than the rest of the left-hand side tokens. We hypothesize that this is to determine which intermediate result to copy from the context.

**Production:** Here is an example that uses 24.51 layers in average:

Please calculate the product of numbers in the following list: [63, 15, 43, 75, 72, 61, 48, 71] **To calculate the product of the numbers in the given list, I'll multiply each number individually**

**Here's the step–by–step calculation**

**1. 63 x 15 = 945**

**2. 945 x 43 = 40515**

**3. 40515 x 75 = 3026125**

**4. 3026125 x 72 = 218171500**

**5. 218171500 x 61 = 13311439500**

**6. 13311439500 x 48 = 637385920000**

**7. 637385920000 x 71 = 45213914128000**

**The final product of the numbers in the provided list is: 45213914128000<|eot_id|>**

Multiplication involves complex iterative computations, as FlexiDepth handles large intermediate results and maintains numerical precision, resulting in higher layer usage. The right-hand side of each equation (e.g., "945", "40515") consistently uses the full number of layers to compute the product, reflecting the task's high computational demand. Similar to addition, on the left-hand side, initial tokens like "63 × 15" in the first step may use slightly more layers to identify the intermediate result to copy from the context. Subsequent tokens (e.g., "945 × 43", "40515 × 75") exhibit a gradually decreasing trend in layer usage, possibly due to increasing certainty when retrieving intermediate results as the computation progresses.

## B  Throughput

We evaluated the throughput of FlexiDepth compared to the vanilla Llama-3-8B-Instruct (Dubey et al., 2024) across the paper's benchmarks. Experiments were conducted on 8 A6000 GPUs with a batch size of 8 and 5-shot examples. For multi-token generation tasks, we fixed the output length at 5 tokens to ensure consistent comparison. All measurements were obtained using the lm-evaluation-harness toolkit (Gao et al., 2024), with throughput reported as iterations per second. Table 5 presents the results across six benchmarks.

## C  FlexiExit

We extend FlexiDepth's layer-skipping mechanism into an early-exit framework called FlexiExit, which allows tokens to exit at intermediate layers. In FlexiExit, we adapt FlexiDepth's router and adaptor modules so that a token flagged for skipping at a given layer bypasses

| Model | MMLU | Hellaswag | Winogrande | GSM8K | HumanEval | CoQA | Retain % |
|-------|------|-----------|------------|-------|-----------|------|----------|
| Llama-3-8B | 0.673 | 0.706 | 0.744 | 0.679 | 0.299 | 0.784 | 100.0% |
| FlexiExit | 0.676 | 0.741 | 0.756 | 0.681 | 0.220 | 0.787 | 96.9% |
| Skipped | 2.17 | 1.81 | 3.63 | 4.68 | 1.61 | 3.16 | - |

Table 6: Performance and skipping patterns of FlexiExit

the attention modules in all subsequent layers and is processed solely by lightweight adaptors. We applied FlexiExit to the Llama-3-8B-Instruct model, training it under the same conditions as FlexiDepth but with a reduced layer-skipping weight of $\alpha = 3 \times 10^{-4}$. As shown in Table 6, FlexiExit retains the performance of the original Llama-3-8B-Instruct, while skipping an average of 1.61 to 4.68 layers. The current design of the router and adapter remains effective in supporting early exit scenarios.

