# OpenReview forum: "Adaptive Layer-skipping in Pre-trained LLMs"
_colmweb.org/COLM/2025/Conference — COLM 2025_

### Official Review · Reviewer_Anmw · 2025-04-14

**Rating:** 6
**Confidence:** 4
**Ethics Flag:** 1

**Summary:**

This works propose FlexiDepth - a layer-skipping strategy that introduces a supervised router module to adaptively skip transformer layers on a token-level basis. The authors introduce several techniques such as non-linear router, KV-cache computation, auxiliary adapter, that allows them to preserve the model performance while skipping a considerable amount of layers. Evaluated across tasks, FlexiDepth can skip layers without compromising performance, and its skipping strategy aligns with human intuition of generation difficulty.

**Reasons To Accept:**

- The method seems overall novel and sound - many design choices are well motivated and are backed by ablations — for example, introducing an adapter that aligns the representation of skipped state with the latent space of unskipped states and computing the KV Cache for skipped tokens.
- The authors present a surprisingly strong result where FlexiDepth works as a kind of regularization at test-time, sometimes surpassing the performance of model performance without layer skipping. This result is further supported by the fact that additional training of the full model on the same dataset does not lead to comparable performance.

**Reasons To Reject:**

- My biggest concern is that while appealing as a method, the practical utility of this method remains uncertain. Across evaluations, the metric of computational efficiency is the number of layers skipped on average. In reality, due to the existence of FFN skipping and router networks, mere number of skipped layers may not fully represent how efficient the method is compared to baselines. While the authors explain in limitations that theoretical improvement does not translate to existing GPU setups, it is important that readers are informed about the actual differences in estimated flops or wall-clock time to understand the current gap between methods.
- The fundamental question raised in the intro is “How do computational demands vary across the generation of different tokens”. While the proposed method indeed tackles this question, the claim that previous methods on layer skipping overlooks this question seems to be overly strong. For example, confidence-adaptive LM (Schuster et al, 2022, cited by authors) exactly tackles this problem on a token-level basis, albeit it differs from FlexiDepth in that it does not learn any parametric module.
- The method introduces an important hyperparameter alpha, which essentially trades off efficiency vs. quality. Depending on tasks the impact of value could significantly vary, thus the hyperparameter search could be non-trivial without large scale validation. Perhaps a good guidance on the value setting would be beneficial for practical usage of FlexiDepth.

---

> ### Author Response · Authors · 2025-06-02
>
> We sincerely thank the reviewer for the feedback. We will address each concern in detail:
>
> **Q1: Efficiency**
>
> We will explicitly include a table about the wallclock time so that readers are informed about the actual differences. The following table shows throughput (samples/s) evaluated on the paper's benchmarks using 8 A6000 GPUs with batch size 4 (Llama-3-8B-Instruct as the base model):
>
> | Method | MMLU | HellaSwag | Winogrande | GSM8K | HumanEval | CoQA |
> |--------|------|-----------|------------|-------|-----------|------|
> | Original | 33.27 | 11.60 | 26.50 | 0.50 | 0.09 | 1.34 |
> | FlexiDepth | 33.91 | 11.44 | 26.36 | 0.49 | 0.08 | 1.31 |
>
>
> **Q2: Claims About Previous Methods**
>
> Thank you for pointing out that confidence-adaptive LM (Schuster et al., 2022) also tackles this problem on a token-level basis. We will revise our wording and modify the related works accordingly.
>
> **Q3: Hyperparameter Setting**
>
> We agree that providing guidance on the hyperparameter α is crucial for practical usage of FlexiDepth. We will add ablation studies showing how different α values affect the number of layers skipped and performance across our benchmark tasks. Our preliminary results show that α values between 1e-4 and 4e-3 provide a good balance between efficiency and quality for most tasks, with α=1e-3 being a robust default choice. We will include a table showing the relationship between α values, the number of layers skipped, and performance metrics across different task categories to help practitioners select appropriate values.

---

> > ### Comment · Reviewer_Anmw · 2025-06-02
> >
> > Thank you for the response. My questions are addressed and I will keep my positive evaluation of the paper.

---

### Official Review · Reviewer_BFv9 · 2025-05-11

**Rating:** 8
**Confidence:** 4
**Ethics Flag:** 1

**Summary:**

The authors develop a layer skipping router to reduce the number of per token computation during generation.

**Questions To Authors:**

What is effect of layer skipping on perplexity?

**Reasons To Accept:**

This is a very sensible idea and the authors find what I would consider to be the most sensical approach, solving practical problems of kv caching and representation alignment along the way. I am impressed that they achieve 100% performance parity with full models. The paper is easy to understand and overall well written (though there are some minor grammar errors.) I like the interpretability aspect.

**Reasons To Reject:**

Unfortunately the most obvious practical benefit here is to save computation and get better throughput but this is not realized. It would have been nice to see more attempts at making that possible rather than giving up (even failed attempts would have been beneficial to the community to learn about)

The interpretability aspect would have been really exciting to focus on more with visualizations and qualitative examples. Instead it is left to future work.

---

> ### Author Response · Authors · 2025-06-02
>
> We sincerely thank the reviewer for the insightful feedback and positive assessment. We address each point below:
>
> **Q1: Efficiency Issue**
>
> In our current implementation, the wall-clock time improvement is not significant for the decoding stage since memory I/O becomes the bottleneck. Further hardware-oriented improvements are possible, particularly if the skipping operations can be made I/O-aware. We will continue working on this issue since the number of skipped layers seems quite significant. We hope the release of this work and its code can also motivate the community to come up with better hardware-oriented solutions.
>
>
> **Q2: More Interpretability**
>
> We agree that deeper interpretability analysis would strengthen our contribution. We are excited to report that we found the layer skipping patterns of FlexiDepth follow a long-tail distribution. Specifically, we measured the percentage of tokens processed by each transformer layer across the tasks outlined in our paper: text generation (copying, summarization, continuation) and math reasoning (repeating, addition, product).  Layer utilization displays a distinctive bowl-like pattern with high usage in initial and final layers but reduced usage in middle layers. For language tasks, copying exhibits the deepest dip in middle-layer usage, while continuation involves the most middle layers, reflecting varying task complexity. Similarly, in math reasoning, repeating shows the least middle-layer usage, whereas product requires the most. We will add detailed visualizations and analysis of these patterns in our revision.
>
> **Q3: Effect on Perplexity**
>
> We evaluated byte-level perplexity on the WikiText benchmark using lm_evaluation_harness codebase. The original Llama-3-8B-Instruct achieves a perplexity of 1.5367, while FlexiDepth with α=1e-3 penalty achieves 1.5753. The proposed FlexiDepth slightly increases the perplexity.

---

> > ### Comment · Reviewer_BFv9 · 2025-06-03
> >
> > Thank you for your responses, I am excited to see the additional interpretability experiments. I maintain my high score and recommendation for acceptance.

---

### Official Review · Reviewer_upxR · 2025-05-14

**Rating:** 8
**Confidence:** 4
**Ethics Flag:** 1

**Summary:**

The paper introduces FlexiDepth -- an approach to dynamically skip layers in pre-trained large language models, without changing the model parameters. FlexiDepth introduces for each layer a router which decides whether this layer will be skipped for each token. Then, it uses adapter through which to pass the values for the skipped layers. During fine-tuning, the weights of the router and adapter are updated, and the model weights are kept frozen.
The method has been evaluated on several diverse benchmarks and compared to several other layer skipping methods. The results show that FlexiDepth outperforms the other methods, and retains the task performance, and in some cases perform even slightly better than the original model.
The paper also introduces a layer allocation dataset with recorded layers skipped for each token.

**Questions To Authors:**

- How hard is it to integrate the method in practice -- what are the necessary changes that need to be done to a model -- is it a simple plug-and-play, or does it require some specific integration?

Suggestions and clarifications:
- KV cache is only mentioned as a name, but is not explained anywhere in the paper. A short explanation would help make the paper self-contained. It is also mentioned in Figure 3, but not discussed where this figure is referenced.
- In the caption of Figure 1: "The light-to-dark blue gradient represents layer usage from 16 to 32." -> Is this the number of layers? Or the layer number - does it use only one layer? Maybe good to clarify.
- When explaining the router design in Section 2.1, it is not explicitly mentioned that there is one router for each layer, and the input X is the input to this layer, not to the network.

Small:
- line 52: " to match those undergo" maybe has a missed preposition.
- line 131: "leads to significantly degrade model performance" sounds weird, maybe some missed word.

**Reasons To Accept:**

The paper is well-written, clear and easy to follow. The proposed method is sound, the experiments are thorough and show that FlexiDepth outperforms other layer skipping methods. The analysis of the layer allocation on various tasks contributes to understanding of the method.
The paper also discusses the limitations of the method and propose how they can be addressed in future work.
The authors promise to open-source the model and the dataset on acceptance.

**Reasons To Reject:**

I do not really see any reasons to reject, the paper looks good. I have some small suggestions, given in the next field.

---

> ### Author Response · Authors · 2025-06-02
>
> We sincerely thank the reviewer for the thorough review and valuable feedback. We address each question below:
>
> **Q1: Integration Complexity and Practical Implementation**
>
> FlexiDepth is designed as a plug-and-play method that preserves all parameters in the pre-trained model. The integration process is straightforward:
>
> **Add modules:** Insert the router and adapter modules to each target layer (e.g., layers 17-32 for Llama-3-8B-Instruct)
>
> **Freeze original parameters:** Keep all original model parameters frozen
>
> **Train only new components:** Fine-tune only the router and adapter parameters
>
> This minimal modification approach ensures easy integration  with existing models.
>
>
> **Q2: KV Cache Explanation**
>
> We will add a brief explanation of KV Cache in Section 2.2 (Attention Skipping) to make the paper self-contained.
>
>
> **Q3: Figure 1 Caption Clarification**
> We will revise the caption to: "The color gradient represents the number of layers used to generate each token, with light blue indicating 16 layers and dark blue indicating 32 layers."
>
>
> **Q4: Router Design Clarification**
>
> We will explicitly mention in Section 2.1 that there is one router for each layer and X is the input to that specific layer.

---

> > ### Comment · Reviewer_upxR · 2025-06-03
> >
> > Thank you for the response! I am keeping my positive evaluation of the work.

---

### Decision · Program_Chairs · 2025-07-08

**Decision:**

Accept

**Comment:**

The reviewers unanimously find this paper interesting and its claims well-supported by experiment. The methods for adaptive layer skipping are interesting and nicely simple, the experiments are clear, and I think it will spur good discussion at COLM. Few cons to accepting the paper. However, adding my own editorial, I’d like the authors to consider acknowledging the limitations in terms of wall clock time much earlier in the paper – preferably in the introduction – since many readers will be primarily interested in wall clock speedup. Discussing what’s necessary to see wall clock speedups with this method earlier in the paper would improve clarity and may help spur further research that attempts to meet those requirements.